# Evaluating the effectiveness of a group-based resilience intervention versus psychoeducation for emergency responders in England: A randomised controlled trial

**Jennifer Wild**[1]*, **Shama El-Salahi**[2], **Michelle Degli Esposti**[3], **Graham R. Thew**[1,4]

**1** Department of Experimental Psychology, University of Oxford, Oxford, United Kingdom, **2** The Oxford Institute of Clinical Psychology Training and Research, University of Oxford, Oxford, United Kingdom, **3** Department of Social Policy and Intervention, University of Oxford, Oxford, United Kingdom, **4** Oxford Health NHS Foundation Trust, Oxford, United Kingdom

* jennifer.wild@psy.ox.ac.uk

## Abstract

### Background

Emergency responders are routinely exposed to traumatic critical incidents and other occupational stressors that place them at higher risk of mental ill health compared to the general population. There is some evidence to suggest that resilience training may improve emergency responders' wellbeing and related health outcomes. The aim of this study was to evaluate the effectiveness of a tertiary service resilience intervention compared to psychoeducation for improving psychological outcomes among emergency workers.

### Methods

We conducted a multicentre, parallel-group, randomised controlled trial. Minim software was used to randomly allocate police, ambulance, fire, and search and rescue services personnel, who were not suffering from depression or post-traumatic stress disorder, to Mind's group intervention or to online psychoeducation on a 3:1 basis. The resilience intervention was group-based and included stress management and mindfulness tools for reducing stress. It was delivered by trained staff at nine centres across England in six sessions, one per week for six weeks. The comparison intervention was psychoeducation about stress and mental health delivered online, one module per week for six weeks. Primary outcomes were assessed by self-report and included wellbeing, resilience, self-efficacy, problem-solving, social capital, confidence in managing mental health, and number of days off work due to illness. Follow-up was conducted at three months. Blinding of participants, researchers and outcome assessment was not possible due to the type of interventions.

### Results

A total of 430 participants (resilience intervention N = 317; psychoeducation N = 113) were randomised and included in intent-to-treat analyses. Linear Mixed-Effects Models did not

**Data Availability Statement:** The data are held in a public repository. The reference is: Wild, Jennifer (2020), "RCT Resilience Intervention vs

Psychoeducation Emergency Workers", Mendeley Data, V1, doi: 10.17632/y7283fkdtb.1.

**Funding:** Mind www.mind.org.uk, CQR00510, awarded to Dr Jennifer Wild. The funders had no role in study design, data collection and analysis, decision to publish, or preparation of the manuscript.

**Competing interests:** The authors have declared that no competing interests exist.

show a significant difference between the interventions, at either the post-intervention or follow-up time points, on any outcome measure.

## Conclusions

The limited success of this intervention is consistent with the wider literature. Future refinements to the intervention may benefit from targeting predictors of resilience and mental ill health.

## Trial registration

ISRCTN registry, ISRCTN79407277.

## Introduction

Emergency responders are routinely exposed to highly stressful, often traumatic, critical incidents as well as organisational stressors, such as increased workload, staff reductions and reduced access to informal support, that place them at higher risk of mental ill health compared to the general population [1–3]. Whilst they dedicate their lives to improving health and public safety, they are more likely than the general population to suffer from trauma-related psychological disorders, such as posttraumatic stress disorder (PTSD) [4]. A survey conducted by the UK's national mental health charity, Mind, found that 87% of UK-based emergency services staff and volunteers reported high levels of ongoing stress, low mood and poor mental health [5]. Interventions that could improve psychological resilience may improve emergency responder wellbeing and related health outcomes.

The last few decades have seen a surge in the development of interventions aimed to improve resilience in emergency worker populations, with resilience generally being defined as the capacity to maintain wellbeing in response to adversity or stress [6]. Despite widespread use, however, there is conflicting evidence for their efficacy with some resilience interventions demonstrating improvements in wellbeing, sleep or stress symptoms [7–11] whilst others show no significant effects on mental or physical health outcomes [12–17]. Evaluations have typically been hampered by heterogeneity in intervention design, content and outcome measurement, and low methodological quality among studies [18–21]. The majority of trials have evaluated interventions aimed at improving resilience against wait-list rather than an active comparator, making it impossible to determine if improvements in resilience or wellbeing are related to active components of the intervention or to non-specific factors, such as contact with a group. It is unclear whether or not a resilience intervention tailored for emergency workers would fare better than an active alternative and would lead to improvements in emergency responder wellbeing, resilience and related health outcomes.

In 2015, Mind introduced their Blue Light programme, the overall aim of which was to improve the mental health of emergency workers in England. Supported by LIBOR funding from the Cabinet Office, the programme included a number of initiatives, one of which focused on resilience. As part of their pilot phase in the development of this initiative, Mind tailored a group-based resilience intervention for delivery to police, ambulance, fire, and search and rescue workers, which had previously been used in military services and administered to high risk populations, such as new mothers and men at risk of social isolation [22]. The intervention was based on their model of resilience, which posits that improving

wellbeing, social capital and use of psychological coping strategies will improve an individual's resilience. The model incorporates the five ways to wellbeing, a set of evidence-based public health messages, identified by the New Economics Foundation, for improving the mental health and wellbeing of the population [23].

This study is a randomised controlled trial evaluating the effectiveness of Mind's pilot phase resilience intervention for emergency workers compared to accessing psychoeducation about mental health. Overall, randomised controlled trials have found no effect of psychoeducation for reducing psychological symptoms [24] or distress in military personnel [25]. We therefore hypothesised that the resilience intervention would be more effective than psychoeducation in improving resilience, wellbeing, self-efficacy, and social capital, as well as in improving emergency workers' confidence to manage their mental health and reduce days off work due to illness. We hypothesised that neuroticism would predict the degree of change participants would experience in wellbeing, resilience, self-efficacy and social capital.

## Methods

### Design

This study is a two-arm, parallel-group randomised controlled trial conducted in England. Participants were randomly allocated on a 3:1 basis to Mind's group-based resilience intervention or to reading mental health information online. There were no changes to the trial design throughout the study. This paper was written in accordance with CONSORT guidelines [26]. Ethical approval was granted by the Medical Sciences Division Research Ethics Committee at the University of Oxford (1/4/15; ref MS-IDREC-C1-2015-059). The protocol was approved by the funder and the ethics committee prior to recruitment and no changes were made to the protocol at any point during the trial. The trial was registered retrospectively during participant follow-up. The reason for the delay in registering the trial was one of time constraints associated with the priority completion of a number of procedures at the outset to ensure the trial of N = 430 emergency responders, including their follow-up, could be completed within 12 months. The authors confirm that all ongoing and related trials for this intervention are registered, and have all been registered prospectively.

### Participants and recruitment

Recruitment was conducted from May to November 2015 in collaboration with local Mind centres and local emergency services at nine selected sites across England: Andover, Brighton and Hove, Coastal West Sussex, Dudley, Southampton, Birmingham, Oxfordshire, Cambridgeshire, and Peterborough and Fenland. Recruitment methods involved giving talks at emergency service sites, circulating emails, posters, and leaflets, and using social media. Emergency workers were directed to Mind's website where they could sign up for the trial via a link to the registration survey on Qualtrics, a secure online software platform. Participants could read and print a PDF copy of the Participant Information Sheet and pause the registration process to discuss questions with the research assistant over the telephone. If they decided to take part, they were emailed an individualised link where they could log-in, re-read the Participant Information Sheet, and complete a consent form and two short screening questionnaires. Participants were screened for depression and suicidal ideation using the Patient Health Questionnaire 9 (PHQ-9) [27], and for post-traumatic stress using the Post-Traumatic Stress Disorder Checklist for DSM-5 (PCL-5) [28]. They were considered eligible if they scored below 10 on the PHQ-9, below 33 on the PCL-5, and 0 on question nine of the PHQ-9, which assesses suicidal ideation. If participants scored above these cut-off points, they had a telephone call with the researchers to discuss whether their symptoms interfered with their lives and whether they wished

treatment. They were re-included in the study if their symptoms had little impact on their lives and they did not want treatment, otherwise they were excluded and signposted for evidence-based psychological treatment within local Improving Access to Psychological Therapies services. To reduce the risk of participants dropping out, eligible participants were asked to confirm they could commit to a six-week programme before they were randomised.

## Interventions

**Resilience intervention.** The resilience intervention was a six-week, group-based course developed for Mind by Shaun Goodwin, a psychotherapist with expertise in transpersonal counselling, and previously delivered in their work with new mothers and men at risk of social isolation [22]. The intervention included information about mental health and experiential exercises drawn from stress management and mindfulness, with the overarching aim to improve wellbeing and use of adaptive coping strategies, such as social support. Table 1 shows an overview of the weekly content. Homework exercises were set between each session to reinforce learning. Each group session lasted 2.5 hours. Mind facilitators attended a one-day workshop on how to deliver the intervention and then weekly supervision whilst it was ongoing.

**Psychoeducation.** The comparison intervention included psychoeducation about six topics: sleep, stress, depression, anger, mindfulness, and post-traumatic stress disorder. These were selected from a range freely available online from Mind's website https://www.mind.org.uk/information-support/types-of-mental-health-problems/, which the researchers then tailored for emergency workers. Each topic was delivered as an online module, one released each week for six weeks during the same six-week period that the resilience intervention took place. Participants completed the modules remotely at a time during the week that suited them. They could contact the research assistant by email if they had any questions about any part of the modules.

## Primary outcome measures

We adopted a liberal approach to primary outcome, registering seven primary outcome measures. This was for the purpose of ensuring there would be no missed effects linked to the intervention. All primary and secondary outcome measures were administered three times during the study: at baseline, post-intervention, and at three-month follow-up. All outcome measures were self-reported assessments. There were no changes to outcome measures after the trial commenced. The Warwick Edinburgh Mental Wellbeing scale [29] assessed wellbeing. Internal reliability for the scale in the sample was excellent; Cronbach's alpha = 0.94. The

**Table 1. Overview of the weekly content of the resilience intervention.**

| Session | Content |
|---|---|
| 1 | Hopes and Expectations. Looking at how stress affects thoughts, feelings, physical wellbeing and behaviour. |
| 2 | Understanding anxiety and learning why we react the way we do. Identifying distorted thoughts and moods. |
| 3 | How we can limit ourselves through habitual negative thoughts and moods. Challenging distorted negative thoughts and moods. |
| 4 | Managing worry. Managing stress. 'Time for me' and learning how to relax and the importance of doing so. Breathing techniques. Controlling panic. |
| 5 | Setting goals and challenges. Understanding passive anger and resistance. Learning about comfort zones and panic zones. |
| 6 | Reviewing learning. Planning for the future. |
| Throughout the course | A different relaxation technique is introduced in each session, including techniques based on mindfulness. |

Connor-Davidson Resilience Scale [5] measured resilience. Internal reliability was excellent; Cronbach's alpha = 0.93. The General Self-Efficacy Scale [30] is a 10-item scale that assessed optimistic self-beliefs for coping with a variety of difficult demands in life. Internal reliability was good; Cronbach's alpha = 0.89. Two questionnaires were administered to assess social capital: the Social Participation scale [31] and the Social Support scale adapted from Sarason et al's scale [32], which has two subscales, Social Support (Home) and Social Support (Work). Internal reliability for the Social Participation scale was excellent: Cronbach's alpha = 0.92. The Social Support scales showed good internal reliability: Social Support (Home) Cronbach's alpha = 0.77, and Social Support (Work) Cronbach's alpha = 0.83. A one-item measure assessed the degree to which participants felt confident to manage their mental health on a scale from 1 = Totally disagree to 7 = Totally Agree. Higher scores reflect greater confidence in managing mental health. We also administered a two-item questionnaire to assess how many days off work due to illness an individual had taken in the past three months (when administered at baseline and follow-up) and past six weeks when administered at post-intervention.

## Secondary outcome measures

The Depressive Attributions Questionnaire [33] assesses attributions of negative events. Internal reliability was excellent; Cronbach's alpha = 0.93. We administered three subscales of the Brief Coping Behaviour Questionnaire [34] to assess adaptive coping (active coping, use of emotional support, and acceptance) and five subscales to assess dysfunctional coping (self-distraction, denial, substance use, self-blame, behavioural disengagement). To the dysfunctional coping subscale, we added wishful thinking, which had previously been shown to correlate with severe stress in paramedics [35]. Internal reliability for the adaptive coping scale was good (Cronbach's alpha = 0.81) and excellent for the dysfunctional scale (Cronbach's alpha = 0.93). The Responses to Intrusions Questionnaire [36] assessed suppression, rumination and intentional numbing in response to stressful events. Internal reliability for each scale was good; suppression, Cronbach's alpha = 0.84; rumination, Cronbach's alpha = 0.90; and intentional numbing, Cronbach's alpha = 0.74. The Ruminative Responses Scale [36] measured the frequency of engaging in dwelling. Internal reliability was excellent; Cronbach's alpha = 0.95. An unpublished trauma screener, adapted from the Clinician Administered PTSD Scale for DSM-5 (CAPS-5) to include events relevant to emergency personnel, was used to record exposure to traumatic events [37]. The PCL-5 was administered to assess symptoms of PTSD [28]. The PHQ-9 [27] assessed severity of depression symptoms. Internal reliability of the scale was good, Cronbach's alpha = 0.86. The General Anxiety Disorder 7 (GAD-7) [38] assessed anxiety. Internal reliability was good, Cronbach's alpha = 0.88. The Alcohol Use Disorders Identification Test [39] measured a person's weekly intake of alcohol and substances and whether it has caused problems for them. Internal reliability was good, Cronbach's alpha = 0.74. We administered an unpublished questionnaire to assess problem solving, which had been used in previous evaluations of Mind's resilience intervention [22]. The questionnaire consisted of eight items to assess a person's perception of how well they can solve problems and achieve goals. Internal reliability in the sample was excellent; Cronbach's alpha = 0.90. The neuroticism subscale of the Short-Form Revised Eysenck Personality Questionnaire [40] assessed emotionality. Internal reliability of this scale was good, Cronbach's alpha = 0.84.

## Perceived helpfulness and adherence assessment

To measure perceived helpfulness of the interventions, participants completed a helpfulness rating, indicating on a scale of 0 to 100% how helpful they found their interventions to be. To

assess facilitator adherence to the resilience protocol, we created a short questionnaire that related to the core elements for each of the six sessions. Each group session was recorded using SanDisk MP3 players and the researchers rated 10% of the sessions for adherence to protocol.

## Sample size

Guidelines set by the Cabinet Office for this study suggested a target sample size of 430. We conducted a power analysis to confirm the sample would be large enough to detect an effect should one exist. We referred to a study by Kuehl et al. [8] who compared a group-based 12-week stress management intervention for police officers against standard practice. The intervention led to between-group improvements in wellbeing with small effect (d = 0.34). A sample size calculation was performed for a superiority trial with continuous outcome. Using an alpha of 0.05, 90% power, a standard deviation for the CD-RISC of 14, and a group differ-ence of 5 points (which equates to $d$ = 0.34 from the previous study), would require a total of 330 participants. Allowing for 20% attrition, a total sample size of N = 398 would be required, suggesting that the target sample size was large enough to detect an effect.

## Randomisation

Participants self-enrolled online, were screened and gave consent. They completed their base-line measures before they were randomised on a 3:1 basis to the resilience intervention or to online psychoeducation. The researchers used Minim software to randomly allocate partici-pants by method of minimisation, stratifying the allocation by site and gender. The research assistant entered eligible participants into Minim and then emailed the allocation result to each participant. This randomisation method allowed allocation concealment to be main-tained and reduced the risk of selection bias. Blinding was not possible due to the type of interventions.

## Procedure

The period from the online screening to the onset of the interventions ranged from a few days to 8 weeks, with the majority of participants beginning their intervention within two weeks. Participants (N = 33) who waited more than four weeks to start their interventions re-com-pleted baseline questionnaires. Participants were contacted by email at post-intervention (6 weeks) and 3 months later with a link to complete follow-up questionnaires. The resilience and psychoeducation interventions were delivered 31 times in four phases from May to December 2015. The mean number of participants per group in the resilience intervention was N = 9. The supporting CONSORT checklist for this trial is available as supporting information. See S1 and S2 Files.

## Statistical methods

Data on the number of sessions/modules completed and perceived helpfulness were analysed descriptively and using one-way ANOVA. Facilitators' adherence to the resilience intervention was assessed through independent ratings of session audio recordings.

Linear Mixed-Effects Models were used for the analysis of the primary and secondary out-come variables. Such models have the advantage of using the available data from all partici-pants who were randomized, as well case as accounting for nested data structures and data missing at random. Time (post-intervention, and three-month follow-up), treatment condi-tion (resilience intervention or online psychoeducation [active control]), and the time-by-con-dition interaction were entered as categorical fixed factors along with the stratification

variables of gender and site. Baseline score was included as a covariate, and a random effect of participant was specified to account for between-person variation. Scores on the primary or secondary outcome measure being evaluated were used as the dependent variable. When analysing secondary outcome measures, the baseline scores of the primary outcome measures were included as additional covariates. All models were estimated using restricted maximum likelihood estimation. Q-Q plots indicated that the normality of residuals assumption was met for all models.

Between-group effect sizes ($d_{\text{Cohen}}$) were calculated by dividing the adjusted group difference by the baseline standard deviation of the full study sample. Within-group effect sizes were calculated from separate models that incorporated the baseline score as a timepoint rather than as a covariate, to permit calculation of within-group changes from baseline. These models used an unstructured covariance matrix. 95% confidence intervals for $d_{\text{Cohen}}$ were calculated by dividing the upper and lower limits of the adjusted group difference by the baseline standard deviation of the full study sample.

A series of linear regressions was performed to examine if baseline neuroticism scores predicted the extent of Baseline-Post change in Wellbeing, Resilience, Self-efficacy, and Social Capital within the treatment group. Residualised gain scores (which represent participants' observed change in relation to that predicted from the overall Baseline-Post relationship) were used as the dependent variable in each analysis, and Gender, Site, and Baseline score were included as covariates.

All analyses used the intention to treat sample and a significance level of 0.05. Analyses were conducted in R version 3.5.1 (R Core Team, 2017) [41]. The packages 'tidyverse' (Wickham, 2017) [42], 'nlme' (Pinheiro, Bates, DebRoy, Sarkar, & R Core Team, 2018) [43], 'jmv' [44] and 'psych' [45] were used. The Confidence in Managing Mental Health variable was log transformed prior to analysis given non-normality of the raw data. The total score for days off work due to illness was non-normal and could not be corrected with transformations. We analysed this variable at post-intervention and follow-up with Mann-Whitney U Tests.

## Results

Four hundred and thirty participants (N = 317 resilience intervention, N = 113 online psychoeducation) took part in the trial from May 2015 to March 2016. Fig 1 shows an overview of the number of participants from enrolment to analysis. Follow-up began in July 2015 and continued until March 2016. The trial ended with the end of Mind's pilot year of the Blue Light programme. The majority of participants were female (58.1%), White British (89.7%), police officers (52.3%), and were on average 41 years old (SD = 9.78). Table 2 describes the study sample by intervention arm. Table 3 shows the means and standard deviations of primary outcome measures at baseline, post-intervention and three-month follow-up.

### Sessions/modules completed, perceived helpfulness and adherence assessments

Participants receiving the resilience intervention completed a mean number of 4.11 group sessions (SD = 2.02) whilst those receiving the psychoeducation intervention completed a mean number of 4.71 (SD = 2.01) modules. Participants receiving psychoeducation completed more modules than sessions attended by participants in the resilience intervention (F(1,429) = 7.21, p = 0.008). Participants rated the resilience intervention (84%, SD = 19.49) as significantly more helpful than the psychoeducation intervention (77%, SD = 22.79), (F(1,278) = 9.4, p = 0.002). Thirty audio-recordings from the group sessions (15%) were randomly selected to measure the facilitators' adherence to protocol whilst delivering the resilience intervention.

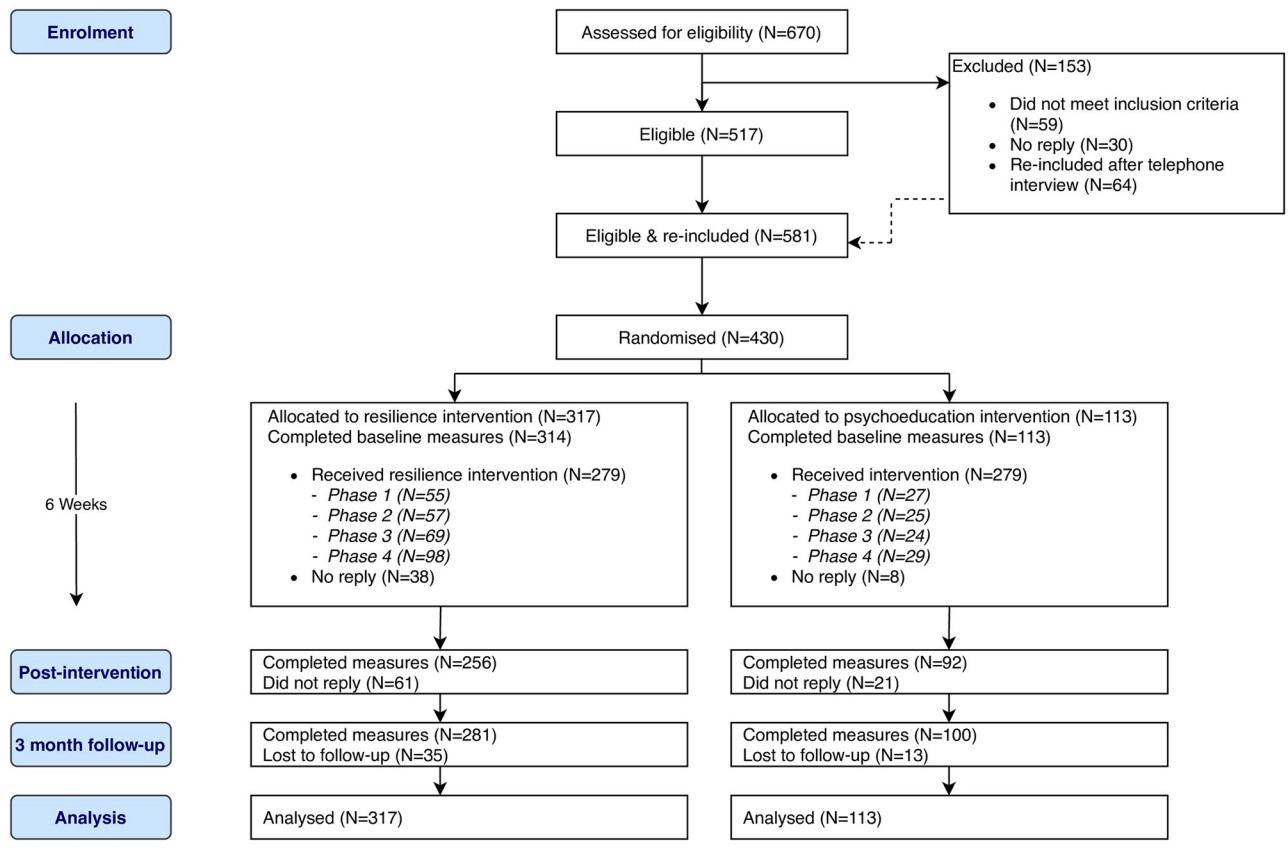

**Fig 1. Consort flow diagram.**

Two research assistants independently rated the recordings for inter-rater reliability, which yielded a correlation coefficient of r = 0.985, suggesting excellent inter-rater reliability. Adherence to protocol ratings ranged from 60% to 100%, with a mean rating of 85.65% (SD = 13.07), suggesting that the facilitators demonstrated good adherence to protocol.

The results of the Linear Mixed-Effects Models are given in Table 4. There were no significant differences between the intervention groups on any of the primary outcome measures at either the post-intervention or three-month follow-up timepoints. The between- and within-group effect sizes suggested there was little to no change on these measures resulting from either intervention. The groups did not differ in the number of days off work due to illness they had taken at post (Mann-Whitney U = 11,684.00, p = 0.892) or at follow-up (Mann-Whitney U = 13,715.00, p = 0.754).

Analysis of the secondary outcome measures showed the same pattern of results, indicating that the resilience intervention was not superior to psychoeducation and did not lead to significant improvements in depressive attributions, coping strategies, responses to intrusions, rumination, or symptoms of PTSD, depression, anxiety, or problematic alcohol use.

Looking at within-subjects effects, participants receiving psychoeducation demonstrated small improvements in wellbeing, social participation, confidence to manage mental health, depressive attributions, dysfunctional coping, rumination and suppression in response to stressful memories at follow-up compared to when they came into the trial. Participants who received the resilience intervention demonstrated small improvements in suppression in response to intrusive memories at post-intervention and at follow-up compared to their

**Table 2. Demographic description of participants at randomisation.**

| | | Resilience Intervention (N = 317) | Psychoeducation (N = 113) | Total (N = 430) |
|---|---|---|---|---|
| **Age** | Mean (SD) | 41.09 (9.98) | 42.32 (9.20) | 41.41 (9.78) |
| **Gender** | Female | 186 (58.68%) | 64 (56.64%) | 250 (58.14%) |
| | Male | 131 (41.32%) | 49 (43.36%) | 180 (41.86%) |
| **Marital Status** | Single | 57 (17.98%) | 19 (16.81%) | 76 (17.67%) |
| | Married | 164 (51.74%) | 51 (45.13%) | 215 (50.00%) |
| | Divorced/Separated | 30 (9.46%) | 14 (12.39%) | 44 (10.23%) |
| | Widowed | 3 (0.95%) | 0 (0%) | 3 (0.70%) |
| | Civil partnership | 3 (0.95%) | 2 (1.78%) | 5 (1.16%) |
| | Long-term partner | 60 (18.93%) | 27 (23.89%) | 87 (20.23%) |
| **Highest Qualification** | GCSE | 56 (17.67%) | 13 (11.50%) | 69 (16.05%) |
| | A-Levels | 82 (25.87%) | 35 (30.97%) | 117 (27.21%) |
| | Degree/College | 140 (44.16%) | 50 (44.25%) | 190 (44.19%) |
| | Masters | 33 (10.41%) | 10 (8.85%) | 43 (10.00%) |
| | PhD or Other qualification | 6 (1.89%) | 5 (4.42%) | 11 (2.56%) |
| **Ethnicity** | White British/European | 299 (94.32%) | 107 (94.69%) | 406 (94.42%) |
| | Black/Indian/Asian/Arab | 18 (5.68%) | 6 (5.31%) | 24 (5.58%) |
| **Service** | Police | 170 (53.63%) | 55 (48.67%) | 225 (52.33%) |
| | Ambulance | 89 (28.08%) | 31 (27.43) | 120 (27.91%) |
| | Fire | 47 (14.83%) | 21 (18.58) | 68 (15.81%) |
| | Search and rescue | 11 (3.47%) | 6 (5.31%) | 17 (3.95%) |

baseline assessment. Model results and descriptive statistics for the secondary outcomes are provided in the supplementary material.

To examine the hypothesis that baseline levels of neuroticism may predict the extent of individual pre-post change in the resilience intervention, a series of linear regressions was conducted using residualised gain scores of the primary outcome measures as the dependent variables. The results (see Table 6 in S3 File) showed that none of the overall models were significant, with low $R^2$ values, indicating there was no evidence within this sample that

**Table 3. Primary outcome measures at baseline, post-intervention and follow-up.**

| | Resilience Intervention | | | Psychoeducation | | |
|---|---|---|---|---|---|---|
| | Baseline (N = 314) | Post (N = 256) | Follow-up (N = 281) | Baseline (N = 113) | Post (N = 92) | Follow-up (N = 100) |
| | Mean (SD) | Mean (SD) | Mean (SD) | Mean (SD) | Mean (SD) | Mean (SD) |
| Resilience (CD-RISC) | 66.49 (14.72) | 67.94 (17.01) | 68.52 (16.18) | 67.48 (14.62) | 68.48 (15.26) | 69.43 (15.25) |
| Wellbeing (WEMWBS) | 48.57 (8.90) | 50.69 (9.36) | 50.29 (9.10) | 48.49 (9.17) | 51.28 (9.93) | 50.76 (9.51) |
| Self-Efficacy (GSE) | 30.94 (4.22) | 31.75 (4.48) | 31.82 (4.58) | 31.69 (4.22) | 31.91 (4.73) | 32.33 (4.54) |
| Social Participation | 59.06 (15.84) | 62.38 (17.82) | 61.38 (16.89) | 56.84 (17.37) | 60.63 (17.90) | 59.94 (18.94) |
| Social Support (Home) | 33.04 (6.08) | 33.64 (6.43) | 34.14 (6.71) | 32.58 (6.86) | 32.83 (7.09) | 33.09 (7.93) |
| Social Support (Work) | 27.20 (6.64) | 27.17 (6.58) | 27.42 (6.80) | 26.51 (6.64) | 27.14 (7.16) | 26.71 (7.13) |
| Days off work/week | 0.25 (0.93) | 0.25 (0.94) | 0.38 (1.42) | 0.28 (1.00) | 0.24 (0.80) | 0.44 (1.29) |
| Confidence to manage mental health | 5.04 (1.32) | 5.42 (1.18) | 5.41 (1.31) | 4.98 (1.43) | 5.42 (1.21) | 5.49 (1.31) |

Notes. Data completeness in the Resilience Intervention group was 99% at baseline, 81% at Post-intervention, and 89% at Three-month Follow-up. In the Psychoeducation group the figures were 100%, 81%, and 88%, respectively. CD-RISC = Connor-Davidson Resilience Scale; WEMWBS = Warwick Edinburgh Mental Wellbeing scale; GSE = General Self-Efficacy.

**Table 4. Adjusted group differences and effect sizes of the primary outcome measures for the intention to treat sample.**

| | Adjusted group difference (*SE*) [95%CI], *p* value | | Effect size $d_{Cohen}$ [95%CI] | | | | |
|---|---|---|---|---|---|---|---|
| | Post | FU | Between-group at Post | Between-group at FU | | Within-group pre-post[a] | Within-group pre-FU[a] |
| WEMWBS | | | | | | | |
| Resilience Intervention vs Psychoeducation | -0.27 (0.85) [-1.94, 1.40], .755 | -0.52 (0.83) [-2.15, 1.11], .532 | 0.03 [-0.16, 0.22] | 0.06 [-0.12, 0.24] | Resilience | 0.03 [-0.16, 0.23] | 0.06 [-0.13, 0.25] |
| | | | | | Psychoeducation | 0.25 [0.09, 0.42] | 0.24 [0.08, 0.40] |
| CDRISC | | | | | | | |
| Resilience Intervention vs Psychoeducation | 0.52 (1.41) [-2.25, 3.29], .712 | -0.44 (1.36) [-3.11, 2.23], .749 | 0.04 [-0.15, 0.22] | 0.03 [-0.15, 0.21] | Resilience | 0.06 [-0.13, 0.26] | <0.01 [-0.19, 0.19] |
| | | | | | Psychoeducation | 0.03 [-0.13, 0.20] | 0.13 [-0.03, 0.30] |
| GSES | | | | | | | |
| Resilience Intervention vs Psychoeducation | 0.55 (0.43) [-0.30, 1.40], .209 | 0.05 (0.42) [-0.78, 0.88], .902 | 0.13 [-0.07, 0.33] | 0.01 [-0.18, 0.21] | Resilience | 0.19 [-0.02, 0.40] | 0.08 [-0.13, 0.28] |
| | | | | | Psychoeducation | 0.01 [-0.18, 0.18] | 0.14 [-0.04, 0.31] |
| Problem Solving | | | | | | | |
| Resilience Intervention vs Psychoeducation | -0.15 (0.47) [-1.07, 0.77], .751 | 0.04 (0.46) [-0.86, 0.94], .929 | 0.03 [-0.16, 0.22] | 0.01 [-0.18, 0.20] | Resilience | 0.01 [-0.20, 0.22] | 0.05 [-0.16, 0.25] |
| | | | | | Psychoeducation | 0.19 [0.02, 0.36] | 0.19 [0.03, 0.36] |
| SPS | | | | | | | |
| Resilience Intervention vs Psychoeducation | 0.67 (1.55) [-2.38, 3.72], .667 | -0.33 (1.51) [-3.30, 2.64], .828 | 0.04 [-0.15, 0.23] | 0.02 [-0.16, 0.20] | Resilience | 0.01 [-0.18, 0.21] | 0.05 [-0.14, 0.24] |
| | | | | | Psychoeducation | 0.20 [0.05, 0.35] | 0.17 [0.03, 0.32] |
| SS(Home) | | | | | | | |
| Resilience Intervention vs Psychoeducation | 0.72 (0.62) [-0.50, 1.94], .246 | 0.60 (0.60) [-0.58, 1.78], .321 | 0.11 [-0.08, 0.31] | 0.10 [-0.09, 0.28] | Resilience | 0.09 [-0.12, 0.30] | 0.08 [-0.12, 0.28] |
| | | | | | Psychoeducation | 0.03 [-0.13, 0.18] | 0.09 [-0.07, 0.24] |
| SS(Work) | | | | | | | |
| Resilience Intervention vs Psychoeducation | -0.41 (0.57) [-1.53, 0.71], .478 | 0.01 (0.55) [-1.07, 1.09], .985 | 0.06 [-0.11, 0.23] | <0.01 [-0.16, 0.16] | Resilience | 0.08 [-0.10, 0.25] | 0.03 [-0.14, 0.20] |
| | | | | | Psychoeducation | 0.07 [-0.08, 0.22] | 0.05 [-0.10, 0.20] |
| CMH | | | | | | | |
| Resilience Intervention vs Psychoeducation | <0.01 (0.01) [-0.02, 0.02], .876 | -0.01 (0.01) [-0.03, 0.01], .522 | <0.01 [-0.14, 0.14] | 0.07 [-0.07, 0.21] | Resilience | 0.04 [-0.21, 0.29] | 0.09 [-0.14, 0.29] |

(*Continued*)

**Table 4.** (Continued)

| | Adjusted group difference (*SE*) | | Effect size $d_{Cohen}$ [95%CI] | | | | |
|---|---|---|---|---|---|---|---|
| | [95%CI], *p* value | | | | | | |
| | Post | FU | Between-group at Post | Between-group at FU | | Within-group pre-post[a] | Within-group pre-FU[a] |
| | | | | | Psychoeducation | 0.29 | 0.30 |
| | | | | | | [0.07, 0.47] | [0.13, 0.47] |

Note. In the Intervention group, 306 participants provided data at baseline, 256 at posttreatment, and 282 at follow-up. In the Psychoeducation group, 108 participants provided data at baseline, 92 at posttreatment, and 100 at three-month follow-up. All Linear Mixed-Effects Models included the baseline score, gender, and site as covariates, and a random effect of participant. WEMWBS = Warwick Edinburgh Mental Wellbeing Scale; CDRISC = Connor-Davidson Resilience Scale; GSES = General Self-Efficacy Scale; SPS = Social Participation Scale; SS = Social Support; CMH = Confidence in Managing Mental Health and Resilience Scale.
[a] Within-group effect sizes obtained from separate Linear Mixed-Effects Models including baseline score as a timepoint (see Method).

neuroticism predicted the extent of change in wellbeing, resilience, self-efficacy, or social capital associated with the intervention.

## Discussion

This randomised controlled trial evaluated the effectiveness of a tertiary service resilience intervention for improving psychological outcomes among emergency workers compared to a psychoeducation-only intervention. There were no significant differences between the interventions on any of the primary or secondary outcome measures at the post-intervention or follow-up timepoints although participants receiving the resilience intervention rated it as more helpful than those receiving psychoeducation.

The results of this trial are consistent with findings that resilience interventions may have limited effects on mental health outcomes in emergency workers [12–17] and the growing concern in the field that although some interventions may improve wellbeing [7–11], it remains to be seen whether or not this translates to better mental health outcomes. Interestingly, modest improvements were observed for participants receiving psychoeducation on some outcomes at follow-up compared to their baseline assessments. However, these are likely to be due to non-specific factors, such as contact with a research assistant, since the differences were not found between the groups on the same outcomes. This would be in keeping with the wider literature, which suggests psychoeducation is generally ineffective in terms of building resilience to stress. For example, a cluster randomised controlled trial attempted to directly measure the impact of psychoeducation among new firefighter recruits and evidenced no long-term benefits of psychoeducation in terms of help seeking or symptom levels [46]. Similarly, Sharpley et al. [25] compared Naval and Marine personnel who had and had not received psychoeducation about stress and stress reactions in a briefing session prior to being deployed to the 2003 Iraq war. There was no evidence that pre-deployment psychoeducation reduced subsequent psychological distress after deployment. What is emerging is evidence that it is the type of education that matters: training about the job rather than training in stress management [i.e., 46–51].

The resilience intervention evaluated in this trial included tools to promote mindfulness and manage stress with the aim of fostering wellbeing, psychological coping, and social capital in an attempt to improve overall resilience. Perhaps a more theory-driven approach to resilience-building is needed, such as identifying and then targeting predictors of resilience and also mental ill health. A recent systematic review of interventions aimed to improve wellbeing

and resilience to stress among emergency responders found that those most likely to demonstrate intervention-specific improvements targeted modifiable risk factors of trauma-related psychological disorders, such as PTSD and depression [18]. This approach is echoed in medicine where interventions for building resilience to ill health target modifiable risk factors, such as targeting hypertension to reduce the risk of cardiovascular disease and mortality [52]. Targeting modifiable risk factors for psychological disorders has also been shown to be effective in preventing the development of depression [53]. A similar approach may be effective for populations, such as emergency responders, at risk of developing severe stress reactions like PTSD. To date, there is little prospective research identifying these risk factors, although a study conducted by Wild et al. [4] identified two predictors of poor mental health that could serve as targets for future interventions.

On the whole, the majority of evaluations of resilience interventions are hampered by trials of low methodological quality and comparison to wait-list rather than active comparison conditions, making it difficult to conclude whether or not any improvements are intervention-specific. The current trial overcame these shortfalls by implementing a robust design with a large sample of emergency workers. However, despite the rigorous approach employed in this trial, there are limitations worth considering. First, all outcome measures were self-report, which are subjective and open to bias. Second, there was no wait-list condition. In addition to an active comparison condition, a wait-list condition allows conclusions to be drawn about intervention-specific effects rather than natural fluctuations in outcomes over time. Third, the interventions differed in their mode of delivery, which may have advantaged one over the other. However, since no effects were found, this is unlikely. Fourth, the follow-up period was fairly short. Fifth, consistent with the model of resilience used to design the intervention, resilience was assessed as a combination of thoughts and behaviours reflective of resilient functioning. Perhaps an operationalised definition of resilience is needed that allows an assessment of better than expected outcomes following stress exposure. Future research could overcome these limitations by including objective assessment measures, such as clinical interviews, including a wait-list arm, and a longer follow-up period that measures exposure to stressful events and subsequent trajectories of outcome.

Notwithstanding the limitations mentioned above, this trial is an important step forward in the development of resilience interventions for emergency workers. It is the first randomised controlled trial to rigorously evaluate a resilience-building programme delivered to a combined sample of emergency responders (i.e., police, paramedics, firefighters and search and rescue personnel) rather than responders from a single service (i.e., police-only). The results are thought to have good generalisability since the intervention was implemented to male and female emergency workers from varied emergency services covering city and rural locations. Our trial demonstrated that a resilience intervention is acceptable to emergency workers and despite their demanding schedules, many were able to commit to a six-week group course. Future resilience interventions may benefit from being tailored to target predictors of resilience and mental ill health in this population.

## Conclusion

We evaluated a tertiary service resilience intervention for emergency workers in a large-scale randomised controlled trial. Although the intervention was acceptable to emergency workers, the results demonstrated that it could not be linked to any intervention-specific improvements in health and wellbeing outcomes. Equally, the comparison condition, psychoeducation, could not be linked to intervention-specific improvements, although participants did fare better on some outcomes at the end of the follow-up period compared to their baseline assessments.

Overall, the resilience intervention performed similarly to psychoeducation, suggesting that it fails to be cost-effective in its current form. The limited success of this intervention is consistent with the wider literature. A more promising approach to developing interventions to improve resilience to stress may be to identify then target modifiable risk factors of stress-related psychopathology.

## Supporting information

**S1 File. CONSORT checklist.**
(DOC)

**S2 File. Trial protocol.**
(PDF)

**S3 File. Secondary outcome.**
(DOCX)

## Acknowledgments

We would like to express our enormous gratitude to the 430 emergency service personnel who took part in this evaluation and the local Mind facilitators who expertly delivered the intervention. We would also like to express our gratitude to the staff at local Minds and at National Mind and the University of Oxford who made this 11-month evaluation possible. In particular, we would like to thank Professor Anke Ehlers, Libby Rackham, Juliane Sachshal, John Fresen, Alice Wallace, Joanna Moss, Shaun Goodwin, Ruth McConkey, Robyn Guillaume-Smith, Krithika Subramanian, Stuart Reid, Gavin Atkins, Faye McGuinness, and Jacob Diggle. The views expressed are those of the authors and not necessarily those of the NHS, the NIHR or the Department of Health.

## Author Contributions

**Conceptualization:** Jennifer Wild.

**Formal analysis:** Jennifer Wild, Graham R. Thew.

**Funding acquisition:** Jennifer Wild.

**Investigation:** Jennifer Wild, Michelle Degli Esposti.

**Methodology:** Jennifer Wild.

**Project administration:** Michelle Degli Esposti.

**Supervision:** Jennifer Wild.

**Visualization:** Shama El-Salahi.

**Writing – original draft:** Jennifer Wild.

**Writing – review & editing:** Jennifer Wild, Shama El-Salahi, Michelle Degli Esposti, Graham R. Thew.

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
