## [Decision Letter · Decision Letter 0]

9 Jun 2020

PONE-D-20-05224

Evaluating the effectiveness of a group-based resilience intervention versus psychoeducation for emergency responders in England: A randomised controlled trial

PLOS ONE

Dear Dr. Wild,

Thank you for submitting your manuscript to PLOS ONE. After careful consideration, we feel that it has merit but does not fully meet PLOS ONE’s publication criteria as it currently stands. Therefore, we invite you to submit a revised version of the manuscript that addresses the points raised during the review process.

We look forward to receiving your revised manuscript.

Kind regards,

Yutaka J. Matsuoka, MD, PhD

Academic Editor

PLOS ONE

Journal Requirements:

3. Thank you for submitting your clinical trial to PLOS ONE and for providing the name of the registry and the registration number. The information in the registry entry suggests that your trial was registered after patient recruitment began. PLOS ONE strongly encourages authors to register all trials before recruiting the first participant in a study.

1) your reasons for your delay in registering this study (after enrolment of participants started);

2) confirmation that all related trials are registered by stating: “The authors confirm that all ongoing and related trials for this drug/intervention are registered”.

Please also ensure you report the date at which the ethics committee approved the study as well as the complete date range for patient recruitment and follow-up in the Methods section of your manuscript.

Additional Editor Comments (if provided):

Reviewers' comments:

Reviewer's Responses to Questions

**Comments to the Author**

1. Is the manuscript technically sound, and do the data support the conclusions?

Reviewer #1: Partly

Reviewer #2: Partly

2. Has the statistical analysis been performed appropriately and rigorously? 

Reviewer #1: No

Reviewer #2: No

3. Have the authors made all data underlying the findings in their manuscript fully available?

Reviewer #1: Yes

Reviewer #2: No

4. Is the manuscript presented in an intelligible fashion and written in standard English?

Reviewer #1: Yes

Reviewer #2: Yes

5. Review Comments to the Author

Reviewer #1: The manuscript entitled ‘Evaluating the effectiveness of a group-based resilience intervention versus psychoeducation for emergency responders in England: A randomised controlled trial’ with the aim to evaluate the effectiveness of a tertiary service resilience intervention compared to psychoeducation for improving psychological outcomes among emergency workers.

The manuscript can be further improved based on the following comments.

Comments

Abstract, the sentence ‘ to the interventions on a 3:1 ratio’ incomplete.

Methods

The mode of administration of all the questionnaires/inventories to be clearly stated. E.g. self-administered or filled up by interviewer/assessor.

Page 11 Line 234 the sentence ‘effect size f=0.17) between two groups (three measurement points: pre-intervention, post-intervention, and follow-up) not clear and more information to be added.

Line 167, for the ‘baseline, Intervention and 3 months’ the period for the intervention to be stated. In some part, post intervention was used. This needs to be standardized. Also could explore the use of symbol T0, T1, T2 to denote the period.

Statistical analyses

Page 12 Line 259, Linear mixed effect models to be written as Linear Mixed-Effects Models.

All statistical tests highlighted in the results section to be stated in the statistical analyses section in the methodology.

Page 12 Line 271, 272, 276, what group differences to be stated.

The level of accepted significance to be stated.

Results

Table 2, for ethnicity White British/European, percentage figures were missing. The highest qualification for resilience intervention has a total percentage 100.2 (if can't be avoided is fine). Total marital status does not tally 100% (the percentage for 87 is incorrect (should be 20.2%). Decimal point to be standardized for all percentage figures. Likewise for the percentage figures in the text.

Table 3, 5 N to be stated on the table.

Information on the dropout rates in % at various point of assessment to be provided.

Table 4, for the data under adjusted difference, denote clearly what data in the Post and FU refers to.

Table 5, some of the SDs are larger than mean. Please check if median ± IQR to be used.

Table 6, figures or parameter indicator to be centralized. R square to be added into discussion to support Page 21 Line 363.

The analysis was based on intent to treat. Were the results any different to per protocol analysis?

Figure 1, baseline to be incorporated in. Post intervention period to be stated.

References did not conform to the journal format.

Reviewer #2: Paper description:

Authors performed a methodologically well balanced study in which they compared effects of two resilience intervention types, aimed for emergency workers: resilience Intervention and psychoeducation. Study is performed on 430 participants, and several primary and secondary outcome measures. Results did not reveal any significant difference between the two intervention types. Also, both interventions showed weak or no effects on primary and secondary outcome measures. Study is well designed and data analysis are well performed.

General comments:

Main shortcoming of this paper is not enough elaborated hypothesis that resilience intervention would be more effective than psychoeducation. Especially having in mind that study did not reveal any significant differences authors must explain in more details why is important to look for such differences. This paper is confirming null hypothesis, and therefore we must have really good argumentation for doing this research.

Another shortcoming is due to sample and analysis. Authors should explain why they used 3:1 ratio for compared groups. In the analysis part, it seems like it is not explicitly mentioned what is used as a dependent variable in mixed models. It seems like authors performed series of analysis in which each score on follow up was predicted by the same score on the baseline (covariate) and other factors. If my assumption is true, then it is hard to expect to get any other significant effects besides the effect of a covariate – since one measure predicts itself best, and does not leave room for other predictors. I would suggest to authors to use differences of baseline-post measures (gains) as dependent variables instead, and to test the effects of factors on those gains. If not instead, then authors should add it as another approach to data analyzing.

Detailed comments:

Abstract: on a 3:1 ratio – please add that it refers to group-based resilience intervention and psychoeducation

“We hypothesised that the resilience intervention would be more effective than psychoeducation in improving resilience, wellbeing” – on what bases is this assumed? Why should we expect this? This must be explained and argumented in more details, especially having in mind that result confirmed null hypothesis.

“...risk of social isolation (Robinson et al., 2014)” – why is this reference in a different formatting?

Did the whole group 314 people attend the session at the same time? Is it too big?

Is it checked if homework exercises were done regularly and how long?

“Psychoeducation about stress and mental health delivered online” – why is one intervention performed online? This leaves room for confounding variable, live versus online training. I understand that it is not too important since no effects are found, but it must be elaborated and argumented.

Is it checked and how, if participants really attended online psychoeducation?

“A one-item 180 questionnaire” – I would suggest to call it one-item measure, since questionnaire usually assumes more items

Why 3:1 ratio for intervention groups? Why not 1:1? This must be explained and argumented

“Residualised gain scores were used as the dependent variable in each analysis...” – what are they residualised from? What where the variables used to separate residuals? Why residuals? Why not just gains?

“Time (post-intervention, and three-month follow-up), treatment condition (resilience intervention or online psychoeducation [active control]), and the time by-condition interaction were entered as categorical fixed factors along” – but it does not seem so form the results, a separate analyses are shown for two time points, it does not seem it was added as a factor

“The resilience and 296 psychoeducation interventions were delivered 31 times in four phases from May to December 297 2015. Follow-up began in July 2015 and continued until March 2016.” –Why is this in the results section, why not in procedure part?

Table 2, for White British/European percentage numbers are missing in brackets. Also in some places a sign % is written on others is missing

“Significantly greater than the number of sessions attended...” – do authors by “sessions” refer to group-based resilience intervention? If yes, please write so, since both interventions types can be referred to as sessions.

“...although participants receiving the resilience 373 intervention rated it as more helpful” – please add statistics on this in the results part

Why is not emergency workers group included as a variable in the analysis? Can authors check are there maybe some effects in some of the groups (police, ambulance, fire, and search and rescue services personnel)

It should be explicitly mentioned what is used as a dependent variable in mixed models. It seems to me that authors performed series of analysis in which each score on follow up was predicted by the same score on the baseline (covariate) and other factors. This should be mentioned explicitly.

If my assumption is true, then it is hard to expect to get any other significant effects besides the effect of a covariate – since one measure predicts itself best, and does not leave room for other predictors. I would suggest to authors to use differences of baseline-post measures (gains) as dependent variables instead, and to test the effects of factors on those gains. If not instead, then authors should add it as another approach to data analyzing.

6. PLOS authors have the option to publish the peer review history of their article (what does this mean?). If published, this will include your full peer review and any attached files.

Reviewer #1: No

Reviewer #2: Yes: Oliver Toskovic

---

## [Author Response · Author response to Decision Letter 0]

18 Sep 2020

Dear Professor Matsuoka

Re: Evaluating the effectiveness of a group-based resilience intervention versus psychoeducation for emergency responders in England: A randomised controlled trial

Thank you for reviewing the above paper. We are grateful for your careful readings of the paper, and pleased that you and the reviewers found the paper of interest. Your suggestions were extremely helpful. We believe the presentation of the manuscript has been further strengthened and trust that it is now acceptable for publication. 

For clarity, each of the revisions is outlined below. Changes to the manuscript are shown in quotations below.

Journal Requirements:

>> Thank you for bringing this to our attention. We have now ensured all parts of the manuscript meet PLOS ONE’s style requirements, including those for file naming.

>> We have readied our database for sharing and are awaiting the accession numbers.

3. Thank you for submitting your clinical trial to PLOS ONE and for providing the name of the registry and the registration number. The information in the registry entry suggests that your trial was registered after patient recruitment began. PLOS ONE strongly encourages authors to register all trials before recruiting the first participant in a study.

1) your reasons for your delay in registering this study (after enrolment of participants started); 

>>The reason for the delay in registering the trial was one of time constraints and has not been repeated with subsequent and related trials. The practicalities of conducting the trial of N=430 emergency responders, including their follow-up, within 12 months necessitated priority completion of a number of procedures at the outset. This meant that the MSD ethics application, the development of the standard operating procedures for nine sites and steps to initiate recruitment and screen participants took priority in the early phase of the study. The authors confirm that the design of the study was approved by both the funder and the MSD ethics committee and was fixed prior to commencing recruitment and no changes were made to the protocol at any point during the trial. The authors confirm that the trial was registered before follow-up was completed and data analysed. The authors confirm that all ongoing and related trials for this intervention are registered, and have all been registered prospectively. 

We have added to lines 124 to 132 on page 6:

“The protocol was approved by the funder and the ethics committee prior to recruitment and no changes were made to the protocol at any point during the trial. The trial was registered retrospectively during participant follow-up. The reason for the delay in registering the trial was one of time constraints associated with the priority completion of a number of procedures at the outset to ensure the trial of N=430 emergency responders, including their follow-up, could be completed within 12 months. The authors confirm that all ongoing and related trials for this intervention are registered, and have all been registered prospectively.” 

2) confirmation that all related trials are registered by stating: “The authors confirm that all ongoing and related trials for this drug/intervention are registered”.

>>Thank you for this suggestion. We have added it to line 132.

Please also ensure you report the date at which the ethics committee approved the study as well as the complete date range for patient recruitment and follow-up in the Methods section of your manuscript.

>>The Medical Sciences Division ethics committee approved the study on 1 April, 2015. This has been added to line 121. An amendment was made to revise the Participant Information Sheet to provide more detail and add two new contacts, which was approved on 15 May, 2015.

Additional Editor Comments (if provided):

5. Review Comments to the Author

Reviewer #1: The manuscript entitled ‘Evaluating the effectiveness of a group-based resilience intervention versus psychoeducation for emergency responders in England: A randomised controlled trial’ with the aim to evaluate the effectiveness of a tertiary service resilience intervention compared to psychoeducation for improving psychological outcomes among emergency workers.

The manuscript can be further improved based on the following comments.

Comments

Abstract, the sentence ‘ to the interventions on a 3:1 ratio’ incomplete.

>>Thank you for highlighting this. We have updated the sentence on lines 34 to 35 as follows:

“We conducted a multicentre, parallel-group, randomised controlled trial. Minim software was used to randomly allocate police, ambulance, fire, and search and rescue services personnel, who were not suffering from depression or post-traumatic stress disorder, to Mind’s group intervention or to online psychoeducation on a 3:1 basis.” 

Methods

The mode of administration of all the questionnaires/inventories to be clearly stated. E.g. self-administered or filled up by interviewer/assessor.

>>Thank you for raising this. We have added to lines 193:

“All outcome measures were self-reported assessments.”

Page 11 Line 234 the sentence ‘effect size f=0.17) between two groups (three measurement points: pre-intervention, post-intervention, and follow-up) not clear and more information to be added.

>>Thank you for raising the need for further clarification on this. We have amended this section to provide a simpler and clearer description of the sample size calculation. Line 275 to 279 now reads:

“A sample size calculation was performed for a superiority trial with continuous outcome. Using an alpha of 0.05, 90% power, a standard deviation for the CD-RISC of 14, and a group difference of 5 points (which equates to d = 0.34 from the previous study), would require a total of 330 participants. Allowing for 20% attrition, a total sample size of N = 398 would be required, suggesting that the target sample size was large enough to detect an effect.”

Line 167, for the ‘baseline, Intervention and 3 months’ the period for the intervention to be stated. In some part, post intervention was used. This needs to be standardized. Also could explore the use of symbol T0, T1, T2 to denote the period.

>>We have reviewed the manuscript and made changes to ensure the timepoints are referred to consistently throughout (baseline, post-intervention, and follow-up).

Statistical analyses

Page 12 Line 259, Linear mixed effect models to be written as Linear Mixed-Effects Models.

>>We have updated the text accordingly.

All statistical tests highlighted in the results section to be stated in the statistical analyses section in the methodology.

>>Thank you for highlighting this. We have now added details of the analysis of sessions/modules completed, ratings of perceived helpfulness, and facilitators’ adherence to protocol, into the ‘Statistical Methods’ section on page 12 lines 314 to 316. 

Page 12 Line 271, 272, 276, what group differences to be stated.

>>Line 276 (now Line 333) refers to between-group effect sizes, estimating the size of any differences between the two intervention groups. Line 271/272 (now Line 326 to 327) just confirms that the models allowed for variation between individual participants – it does not refer to group differences.

The level of accepted significance to be stated.

>>This is now included in line 346:

“All analyses used the intention to treat sample and a significance level of 0.05.”

Results

Table 2, for ethnicity White British/European, percentage figures were missing. The highest qualification for resilience intervention has a total percentage 100.2 (if can't be avoided is fine). Total marital status does not tally 100% (the percentage for 87 is incorrect (should be 20.2%). Decimal point to be standardized for all percentage figures. Likewise for the percentage figures in the text.

>>Thank you for bringing this to our attention. We have now corrected the figures for ethnicity, qualifications and marital status. We have also standardised the percentage figures to two decimal points for the figure, all tables and throughout the text.

Information on the dropout rates in % at various point of assessment to be provided.

>> We have added information on the percentage of data completion at each timepoint to the notes section of Table 3. In addition, the study flowchart has also been updated to clarify rates of dropout/retention. 

Table 4, for the data under adjusted difference, denote clearly what data in the Post and FU refers to.

>>The heading has been changed to ‘Adjusted group difference’ to highlight that these columns represent the difference between the two groups.

Table 5, some of the SDs are larger than mean. Please check if median ± IQR to be used.

>>It is correct that some of the SD values in this table (for the PCL-5, PHQ-9, and GAD-7) are greater than the mean. Following your suggestion, we have added the median and IQR values in these cases.

Table 6, figures or parameter indicator to be centralized. R square to be added into discussion to support Page 21 Line 363.

>>We have amended the formatting as suggested. The R square values are now highlighted within the discussion section on line 472.

“The results (see Table 6, Supplementary Material) showed that none of the overall models were significant, with low R2 values, indicating there was no evidence within this sample that neuroticism predicted the extent of change in wellbeing, resilience, self-efficacy, or social capital associated with the intervention.”

The analysis was based on intent to treat. Were the results any different to per protocol analysis?

>>The results were not any different for the per protocol analysis. The ITT analyses are presented as they are more conservative. 

References did not conform to the journal format.

>>Thank you for bringing this to our attention. We had formatted the References in Vancouver style and have made corrections to individual references which did not quite conform to this style.

Reviewer #2: Paper description:

Authors performed a methodologically well balanced study in which they compared effects of two resilience intervention types, aimed for emergency workers: resilience Intervention and psychoeducation. Study is performed on 430 participants, and several primary and secondary outcome measures. Results did not reveal any significant difference between the two intervention types. Also, both interventions showed weak or no effects on primary and secondary outcome measures. Study is well designed and data analysis are well performed.

General comments:

Main shortcoming of this paper is not enough elaborated hypothesis that resilience intervention would be more effective than psychoeducation. Especially having in mind that study did not reveal any significant differences authors must explain in more details why is important to look for such differences. This paper is confirming null hypothesis, and therefore we must have really good argumentation for doing this research. – don’t need good argumentation once results are null; why it is important to look at resilience

>>Thank you for raising this point. We review the literature on resilience interventions on p4 and their potential promise for improving resilience to stress. We have now added the following on lines 105 to 108 on p5 and refer to 2 studies, including 1 RCT, which found no effect for psychoeducation in terms of reducing psychological symptoms or distress or in improving help-seeking behaviour among emergency workers and military personnel. This research forms the rationale for the hypothesis.

“Overall, randomised controlled trials have found no effect of psychoeducation for reducing psychological symptoms [24] or distress in military personnel [25]. We therefore hypothesised that the resilience intervention would be more effective than psychoeducation in improving resilience, wellbeing, self-efficacy, and social capital, as well as in improving emergency workers’ confidence to manage their mental health and reduce days off work due to illness. We hypothesised that neuroticism would predict the degree of change participants would experience in wellbeing, resilience, self-efficacy and social capital. Another shortcoming is due to sample and analysis. Authors should explain why they used 3:1 ratio for compared groups.” 

>>Thank you for raising this point. It is common for studies in this field to use weighted randomisation. We chose weighted randomisation so as to provide a greater incentive for emergency responders to take part. Since the study was adequately powered, we expected to see an effect should there be one. 

In the analysis part, it seems like it is not explicitly mentioned what is used as a dependent variable in mixed models. It seems like authors performed series of analysis in which each score on follow up was predicted by the same score on the baseline (covariate) and other factors. If my assumption is true, then it is hard to expect to get any other significant effects besides the effect of a covariate – since one measure predicts itself best, and does not leave room for other predictors. I would suggest to authors to use differences of baseline-post measures (gains) as dependent variables instead, and to test the effects of factors on those gains. If not instead, then authors should add it as another approach to data analyzing.

>>Thank you, we have added a sentence in the Statistical Methods section to clarify the specification of the dependent variable (Line 327):

“Scores on the primary or secondary outcome measure being evaluated were used as the dependent variable.”

You raise an interesting point about the different options that might be considered for the dependent variable. The choice made here was based on guidance from a trial statistician, who advised that using follow-up scores and adjusting for baseline scores is preferable to using a change score as the dependent variable when analysing RCTs. This is largely because this method is better at accounting for chance baseline imbalances between the arms of the trial. We would note that this method does seem to be the current prevailing approach for the analysis of RCT data, and that the results of these suggest it remains possible to observe significant group differences even in the presence of baseline score as a covariate.

Detailed comments:

Abstract: on a 3:1 ratio – please add that it refers to group-based resilience intervention and psychoeducation

>> Thank you, this has been corrected.

“We hypothesised that the resilience intervention would be more effective than psychoeducation in improving resilience, wellbeing” – on what bases is this assumed? Why should we expect this? This must be explained and argumented in more details, especially having in mind that result confirmed null hypothesis.

>> Thank you, we have included the following rationale on page 5 lines 100-106.

“Overall, randomised controlled trials have found no effect of psychoeducation for reducing psychological symptoms [24] or distress in military personnel [25]. We therefore hypothesised that the resilience intervention would be more effective than psychoeducation in improving resilience, wellbeing, self-efficacy, and social capital, as well as in improving emergency workers’ confidence to manage their mental health and reduce days off work due to illness. We hypothesised that neuroticism would predict the degree of change participants would experience in wellbeing, resilience, self-efficacy and social capital. Another shortcoming is due to sample and analysis. Authors should explain why they used 3:1 ratio for compared groups.” 

“...risk of social isolation (Robinson et al., 2014)” – why is this reference in a different formatting?

>> Thank you for spotting this error. We have now corrected this. 

Did the whole group 314 people attend the session at the same time? Is it too big?

Is it checked if homework exercises were done regularly and how long?

>> The resilience intervention was delivered 31 times in four phases from May to December 2015. The mean number of participants per group in the resilience intervention was N=9. We have added this to p12 lines 306-309.

“Psychoeducation about stress and mental health delivered online” – why is one intervention performed online? This leaves room for confounding variable, live versus online training. I understand that it is not too important since no effects are found, but it must be elaborated and argumented. 

>>The resilience intervention was designed by Mind as a group intervention and had previously been delivered as such to new mothers and men at risk of social isolation. Online psychoeducation offered a feasible comparison to the resilience intervention, which was attractive to emergency responders since it avoided the scheduling constraints of in-person group sessions. We were also keen to evaluate any potential effects of online psychoeducation, which can be delivered and accessed more easily than group sessions. 

We appreciate that live training might be considered more engaging and that the modes of intervention delivery differ. We have discussed this as a limitation in the Discussion on p23, lines 530 to 531:

Third, the interventions differed in their mode of delivery, which may have advantaged one over the other. However, since no effects were found, this is unlikely. 

Is it checked and how, if participants really attended online psychoeducation?

>> We checked completion rates for each module. The software with which the online psychoeducation was delivered records when a participant has completed the module. We report this on page 16, line 410: 

Participants receiving psychoeducation completed a mean number of 4.71 (SD=2.01) topics.

“A one-item 180 questionnaire” – I would suggest to call it one-item measure, since questionnaire usually assumes more items

>> Thank you for bringing this to our attention. We have now corrected this. Please see line 208.

Why 3:1 ratio for intervention groups? Why not 1:1? This must be explained and argumented

>>Thank you for raising this point. It is standard in this field to use weighted randomisation. We chose weighted randomisation so as to provide a greater incentive for emergency responders to take part. Since the study was adequately powered, should there be an effect, we would have been able to detect one.

“Residualised gain scores were used as the dependent variable in each analysis...” – what are they residualised from? What where the variables used to separate residuals? Why residuals? Why not just gains?

>>The Residualised gain scores are the residuals of a linear regression to predict Post-intervention scores from baseline scores. They represent the difference between a participant’s Observed score at Post, and the score predicted by the overall Baseline-Post relationship. These values therefore capture the participant’s extent of change over time, scaled in relation to the ‘average’ change. The main advantage of this approach is that it avoids using the raw baseline scores in the final variable – these can introduce bias in that participants with more severe baseline scores are more likely to show greater gains due to regression to the mean.

We have added a definition in the Statistical Methods section (page 13, lines 341 to 343) to clarify this:

“Residualised gain scores (which represent participants’ observed change in relation to that predicted from the overall Baseline-Post relationship) were used as the dependent variable in each analysis…”

“Time (post-intervention, and three-month follow-up), treatment condition (resilience intervention or online psychoeducation [active control]), and the time by-condition interaction were entered as categorical fixed factors along” – but it does not seem so form the results, a separate analyses are shown for two time points, it does not seem it was added as a factor

>>As Time is treated categorically in these models, estimates are generated for the group difference at each time point separately. The Time*Condition interaction is therefore not estimated explicitly (as in ANCOVA), but we can confirm that it was included within the models, and that the estimates for the two time points come from a single model (i.e. they are not separate analyses).

“The resilience and 296 psychoeducation interventions were delivered 31 times in four phases from May to December 297 2015. Follow-up began in July 2015 and continued until March 2016.” –Why is this in the results section, why not in procedure part?

>> Thank you for spotting this. We have now moved this to the Procedure on p12 lines 306-309.

Table 2, for White British/European percentage numbers are missing in brackets. Also in some places a sign % is written on others is missing

>> Thank you for spotting this. We have now corrected this table.

“Significantly greater than the number of sessions attended...” – do authors by “sessions” refer to group-based resilience intervention? If yes, please write so, since both interventions types can be referred to as sessions.

>> Thank you for spotting this. We have now corrected this in the manuscript on p16, line 410.

“...although participants receiving the resilience 373 intervention rated it as more helpful” – 

>> Thank you- the results of these ratings are presented on p16. We have also added the following to the Methods on page 12, lines 313 to 315.

Data on the number of sessions/modules completed and perceived helpfulness were analysed descriptively and using one-way ANOVA. Facilitators’ adherence to the resilience intervention was assessed through independent ratings of session audio recordings.

Why is not emergency workers group included as a variable in the analysis? Can authors check are there maybe some effects in some of the groups (police, ambulance, fire, and search and rescue services personnel) 

>>There are no theory-driven hypotheses to suggest that certain subgroups of emergency responders would respond differently to a resilience intervention compared to others. In line with this, sub-group analyses of this type were not planned or registered as part of this study. We therefore did not run subgroup analyses and we would not expect to see differences among them. 

It should be explicitly mentioned what is used as a dependent variable in mixed models. It seems to me that authors performed series of analysis in which each score on follow up was predicted by the same score on the baseline (covariate) and other factors. This should be mentioned explicitly.

If my assumption is true, then it is hard to expect to get any other significant effects besides the effect of a covariate – since one measure predicts itself best, and does not leave room for other predictors. I would suggest to authors to use differences of baseline-post measures (gains) as dependent variables instead, and to test the effects of factors on those gains. If not instead, then authors should add it as another approach to data analyzing.

>>Please see our response to this comment in the ‘General Points’ section of your review. 

Thank you again for your time and attention. We look forward to hearing from you.

Yours sincerely, 

Dr Jennifer Wild & Dr Graham Thew

---

## [Decision Letter · Decision Letter 1]

9 Oct 2020

PONE-D-20-05224R1

Evaluating the effectiveness of a group-based resilience intervention versus psychoeducation for emergency responders in England: A randomised controlled trial

PLOS ONE

Dear Dr. Wild,

Thank you for submitting your manuscript to PLOS ONE. After careful consideration, we feel that it has merit but does not fully meet PLOS ONE’s publication criteria as it currently stands. Therefore, we invite you to submit a revised version of the manuscript that addresses the points raised during the review process.

I kindly ask the authors to address the points by reviewer. I am waiting for your revision.

We look forward to receiving your revised manuscript.

Kind regards,

Yutaka J. Matsuoka, MD, PhD

Academic Editor

PLOS ONE

Reviewers' comments:

Reviewer's Responses to Questions

**Comments to the Author**

1. If the authors have adequately addressed your comments raised in a previous round of review and you feel that this manuscript is now acceptable for publication, you may indicate that here to bypass the “Comments to the Author” section, enter your conflict of interest statement in the “Confidential to Editor” section, and submit your "Accept" recommendation.

Reviewer #1: All comments have been addressed

Reviewer #2: All comments have been addressed

2. Is the manuscript technically sound, and do the data support the conclusions?

Reviewer #1: Partly

Reviewer #2: Partly

3. Has the statistical analysis been performed appropriately and rigorously? 

Reviewer #1: Yes

Reviewer #2: Yes

4. Have the authors made all data underlying the findings in their manuscript fully available?

Reviewer #1: Yes

Reviewer #2: Yes

5. Is the manuscript presented in an intelligible fashion and written in standard English?

Reviewer #1: Yes

Reviewer #2: Yes

6. Review Comments to the Author

Reviewer #1: (No Response)

Reviewer #2: I would point out that authors made significant improvements in the paper. Now it is much clearer what are analyses referring too and how they can be interpreted, at least to me. However I still have few minor concerns:

1. “The Time*Condition interaction is therefore not estimated explicitly (as in ANCOVA)...” – I would suggest that authors mention this somewhere in the text or so. Maybe I was not reading carefully but since it confused me, I believe it will confuse many readers of the paper.

2. I think it would be useful to mention somewhere in analysis part or maybe in the results part, that effect size measures in some sense represent the other approach to group difference testing. The first one is by Linear Mixed-Effects Models, and the other one (more similar to classical ones) is effect sizes approach (targeting mean differences instead of residuals). I strongly believe that this would help in clarity of the text for many readers.

3. I would suggest showing results for the secondary measures, too, at least in the appendix. It would be easier to see, track and so on. For instance, primary measures effect sizes are shown in table, but vaguely commented, while secondary measures differences are commented in more details, and no table is shown in the text. I think that adding it at least in appendix would make it a bit clearer, since readers might see the data text is referring too.

4. Authors write “...receiving psychoeducation completed a mean number of 328 4.71 (SD=2.01) modules, which was significantly greater than the number of sessions 329 attended (F(1,429)=7.21, p=0.008)”, but it confuse me, since I do not understand how can a number of completed be greater than the number of attended sessions? Maybe I misunderstood it, but I believe that other readers can misunderstand it too, so at least it requires come kind of explanation.

I would like to thank to authors for considering my reviews and changing their their text in accordance to comments.

7. PLOS authors have the option to publish the peer review history of their article (what does this mean?). If published, this will include your full peer review and any attached files.

Reviewer #1: No

Reviewer #2: **Yes: **Oliver Toskovic

---

## [Author Response · Author response to Decision Letter 1]

14 Oct 2020

Dear Professor Matsuoka

Re: Evaluating the effectiveness of a group-based resilience intervention versus psychoeducation for emergency responders in England: A randomised controlled trial

Thank you for reviewing the above paper. We are grateful for you and the reviewers’ careful readings of the paper. We have now incorporated the helpful suggestions of Reviewer 2. We believe the presentation of the manuscript has been further strengthened and trust that it is now acceptable for publication. 

For clarity, each of the revisions is outlined below. Changes to the manuscript are shown in quotations below.

Review Comments to the Author

Reviewer #2: I would point out that authors made significant improvements in the paper. Now it is much clearer what are analyses referring too and how they can be interpreted, at least to me. However I still have few minor concerns:

1. “The Time*Condition interaction is therefore not estimated explicitly (as in ANCOVA)...” – I would suggest that authors mention this somewhere in the text or so. Maybe I was not reading carefully but since it confused me, I believe it will confuse many readers of the paper.

Thank you for this suggestion. We have added the below sentence into the Method in order to clarify that the Time*Condition interaction enables the calculation of the treatment effect at each timepoint. We hope that this makes it clearer for readers that the treatment effect at each timepoint is therefore the main outcome of interest.

“The time-by-condition interaction was also entered as a fixed effect in order to allow the estimation of the treatment effect at each timepoint.”

2. I think it would be useful to mention somewhere in analysis part or maybe in the results part, that effect size measures in some sense represent the other approach to group difference testing. The first one is by Linear Mixed-Effects Models, and the other one (more similar to classical ones) is effect sizes approach (targeting mean differences instead of residuals). I strongly believe that this would help in clarity of the text for many readers.

We have amended the below sentence from the Method to incorporate your suggestion, and provide a clearer rationale for the additional use of Effect sizes.

“Between-group effect sizes (dCohen), based on a mean differences approach, were then calculated from the results of these models in order to assist with interpretation of the findings.”

3. I would suggest showing results for the secondary measures, too, at least in the appendix. It would be easier to see, track and so on. For instance, primary measures effect sizes are shown in table, but vaguely commented, while secondary measures differences are commented in more details, and no table is shown in the text. I think that adding it at least in appendix would make it a bit clearer, since readers might see the data text is referring too.

>>Thank you for this suggestion. All secondary outcomes are shown in Tables 5 and 6 in the Supporting Information files. These were submitted in the previous revision and can be found on the page of the submission that says “Click here to download Supporting Information S2 Secondary Outcome Table 5” and “Click here to download Supporting Information S2 Secondary Outcome Table 6.”

4. Authors write “...receiving psychoeducation completed a mean number of 328 4.71 (SD=2.01) modules, which was significantly greater than the number of sessions 329 attended (F(1,429)=7.21, p=0.008)”, but it confuse me, since I do not understand how can a number of completed be greater than the number of attended sessions? Maybe I misunderstood it, but I believe that other readers can misunderstand it too, so at least it requires come kind of explanation.

>>We agree it would be helpful to clarify this further in the text. The modules formed the psychoeducation condition and group sessions formed the resilience intervention. We have clarified this point on lines 326 to 329:

Participants receiving the resilience intervention completed a mean number of 4.11 group sessions (SD=2.02) whilst those receiving the psychoeducation intervention completed a mean number of 4.71 (SD=2.01) modules. Participants receiving psychoeducation completed more modules than sessions attended by participants in the resilience intervention (F(1,429)=7.21, p=0.008).

Thank you again for your time and attention. We look forward to hearing from you.

Yours sincerely, 

Dr Jennifer Wild & Dr Graham Thew

---

## [Editor Report · Decision Letter 2]

20 Oct 2020

Evaluating the effectiveness of a group-based resilience intervention versus psychoeducation for emergency responders in England: A randomised controlled trial

PONE-D-20-05224R2

Dear Dr. Wild,

We’re pleased to inform you that your manuscript has been judged scientifically suitable for publication and will be formally accepted for publication once it meets all outstanding technical requirements.

Kind regards,

Yutaka J. Matsuoka, MD, PhD

Academic Editor

PLOS ONE
---

## [Editor Report · Acceptance letter]

26 Oct 2020

PONE-D-20-05224R2 

Evaluating the effectiveness of a group-based resilience intervention versus psychoeducation for emergency responders in England: A randomised controlled trial 

Dear Dr. Wild:

I'm pleased to inform you that your manuscript has been deemed suitable for publication in PLOS ONE. Congratulations! Your manuscript is now with our production department. 

Kind regards, 

on behalf of

Dr. Yutaka J. Matsuoka 

Academic Editor

PLOS ONE